# Microplastics as both a driver of genitourinary cancers and a deliverer of treatments
Kannan Sridharan [1], Brigida Anna Maiorano[2], Farah Rehan[3], Francesca Maradonna[4,5], Elisabetta Giorgini[4], Tarek Taha[6], Javier Molina-Cerrillo[7], Sebastiano Buti [8,9], Francesco Piva[10], Francesco Massari [11,12,15] ✉ & Matteo Santoni[13,14,15]

Microplastics and nanoplastics (MNPs) are environmental contaminants increasingly detected in human tissues, raising public health concerns. Although evidence is still insufficient to directly link MNPs to genitourinary cancers (GU), this Review examines their potential role in prostate, bladder, and renal cell carcinoma. Proposed mechanisms include chronic inflammation, oxidative stress, genotoxicity, and endocrine disruption driven by plastic-associated additives. Emerging studies report quantitative detection of MNPs within human prostate and bladder tumors, with higher burdens associated with dietary habits such as frequent take-out food consumption. The Review also highlights their role in cancer therapy: MNPs may alter antineoplastic drug pharmacokinetics and promote resistance, yet polymer-based nanoparticles can be engineered as advanced drug delivery platforms. Materials such as PLGA and PEG may improve targeted delivery of chemotherapies and immunotherapies, supporting more effective and personalized treatment strategies in GU oncology.

Microplastics and nanoplastics (MNPs) are non-biodegradable solid particles composed of synthetic polymers, such as polypropylene (PP), polystyrene (PS), or polyethylene (PE), and chemical additives. Microplastics typically range in size from 1 μm to 5 mm, whereas nanoplastics are smaller particles, with a size <1 μm. Primary MNPs are intentionally manufactured for commercial applications (e.g., microbeads in cosmetics, biomedical products, soaps, and toothpaste). In contrast, secondary MNPs originate from the fragmentation of larger plastic due to environmental degradation and ultraviolet (UV) radiation[1]. The persistence of MNPs in the environment is exacerbated by their resistance to natural degradation and poor water solubility, leading to long-term accumulation in ecosystems[1]. Globally, plastic waste management remains inefficient, with 50% of plastics having a short lifespan, 30% an intermediate lifespan, and 20% enduring for extended periods. Due to high recycling costs, nearly 80% of plastic waste ends up in landfills or is indiscriminately released into the environment,

further contributing to MNP pollution[2]. Once released, MNPs undergo physical and chemical transformations (e.g., UV-induced weathering, adsorption of pollutants) altering their mechanical properties and increasing their potential to interact with biological systems, posing significant risks to both environmental and human health[3]. Their ubiquitous presence has led to widespread contamination of air, water, soil, and the food chain, with documented accidental ingestion across diverse species, including fish, marine worms, seabirds, and ultimately, humans[4]. According to World Health Organization (WHO) guidelines, the safe limit for human exposure to styrene monomers is set at a time-weighted average of 20 parts per million (85 mg/m³), with a maximum short-term exposure of 40 ppm (170 mg/m³)[5]. Furthermore, a range of other hazardous substances used in plastic production are now pervasive environmental pollutants. These include catalysts and monofunctional peroxides (e.g., zeolites and iron (III) oxides), along with stabilizers and emulsifiers such as bis(2,2,6,6-

[1]Department of Pharmacology & Therapeutics, College of Medicine & Health Sciences, Arabian Gulf University, Manama, Kingdom of Bahrain. [2]Department of Medical Oncology, IRCCS San Raffaele Hospital, Milan, Italy. [3]Department of Molecular Medicine, College of Medicine & Health Sciences, Arabian Gulf University, Manama, Kingdom of Bahrain. [4]Department of Life and Environmental Sciences, Università Politecnica delle Marche, Via Brecce Bianche, Ancona, Italy. [5]INBB-Consorzio Interuniversitario di Biosistemi e Biostrutture, Roma, Italy. [6]Royal Marsden NHS Foundation Trust, London, UK. [7]Department of Medical Oncology, Hospital Universitario Ramón y Cajal, Madrid, Spain. [8]Medical Oncology Unit, University Hospital of Parma, Parma, Italy. [9]Department of Medicine and Surgery, University of Parma, Parma, Italy. [10]Department of Specialistic Clinical and Odontostomatological Sciences, Polytechnic University of Marche, Ancona, Italy. [11]Medical Oncology, IRCCS Azienda Ospedaliero-Universitaria di Bologna, Bologna, Italy. [12]Department of Medical and Surgical Sciences (DIMEC), University of Bologna, Bologna, Italy. [13]Medical Oncology Unit, Macerata Hospital, Macerata, Italy. [14]Aron Research Foundation ETS, Macerata, Italy. [15]These authors contributed equally: Francesco Massari, Matteo Santoni. ✉e-mail: francesco.massari8@unibo.it

tetramethylpiperidin-4-yl) decanedioate, which are all involved in creating polystyrene particles. Additionally, chemical initiators such as benzoyl peroxide and azobisisobutyronitrile, employed to speed up the polymerization process, are also found globally as environmental contaminants[6,7].

MNPs could be considered exposure biomarkers because they can be detected analytically, may persist over time, and capture contact with contaminated environments via ingestion, inhalation, or potentially dermal uptake. At the same time, substantial between-person variability, driven by differences in diet, exposure patterns, microbiome profiles, and physiological determinants of translocation and clearance, can produce highly heterogeneous internal burdens. Human exposure to MNPs occurs through multiple pathways, with an estimated average intake of 4.1 µg per week, equivalent to consuming a credit card's worth of plastic annually (or roughly 50 plastic bags per year)[8]. The primary routes of exposure include enteral ingestion via contaminated food and water as shown in Fig. 1, with the highest intake from fruits and vegetables (4.78–5.58 × 10⁵ particles/year), drinking water (2.2–12 × 10⁵ particles/year), and table salt (5–7 × 10³ particles/year), and inhalation of airborne microplastics, particularly in indoor environments (1.6–2.3 × 10⁵ particles/year) compared to outdoor air (0.46–2.1 × 10⁵ particles/year)[9]. Dermal absorption, although less characterized, occurs through personal care products (e.g., soaps, hand washes, sunscreens, and toothpaste), the exact quantity absorbed remain unknown[10]. In 2019, the WHO reported no safety concerns for human health due to the presence of MNPs in drinking water[11]. However, since then, a growing body of evidence has investigated the potential harm of MNPs to various aspects of human health, including cancer. Indeed, MNPs are under investigation for their potential role in the development of various solid tumors, as they are involved in shaping the immune system and tumor microenvironment (TME), as well as in cell growth and proliferation[12]. There is limited evidence regarding the involvement of MNPs in the development of tumors in the genitourinary (GU) tract. However, this group of tumors has a widespread diffusion, with prostate cancer (PCa) being one of the most common cancers in the male population. Moreover, GU tumors are associated with excretory function, which is ultimately involved in metabolizing and eliminating various particles from the human body.

In this Review, we provide an overview of the relationship between MNPs and GU tumors, summarizing the link between MNPs and tumor development. Given their ready take up by the body we also discuss their interaction with cancer therapy, and potential future applications as anti-neoplastic therapeutic agents.

## The fundamental differences between environmental MNPs and engineered polymeric nanoparticles

A rigorous distinction should be made between unintentionally generated environmental micro-/nanoplastics (MNPs) and engineered polymeric nanoparticles designed for biomedical use, since they differ fundamentally in origin, degree of physicochemical control, exposure scenarios, and the strength of the safety evidence available (Table 1). From a physicochemical standpoint, environmental MNPs are intrinsically heterogeneous systems, characterized by mixed polymer composition, variable additive content, broad and polydisperse size distributions extending from the nanoscale to the millimeter scale, and irregular morphologies including fibers, fragments, films, and occasional spheres. Their surface properties are also highly dynamic: UV-driven oxidation, mechanical abrasion, and environmental weathering can modify roughness, hydrophobicity, dispersibility, and surface reactivity, while adsorbed biomolecules, biofilms, metals, and persistent organic pollutants further reshape the particle–biointerface in ways that are difficult to predict experimentally[13,14]. As a result, environmental MNPs typically display unstable and poorly standardized surface chemistry, charge, and contaminant burden, all of which can substantially influence biological interactions, tissue distribution, and toxicological readouts. Human exposure is typically chronic, involuntary, and occurs through multiple routes, most notably ingestion and inhalation. At the same time, estimating internal dose remains difficult because analytical approaches are heterogeneous and prone to contamination artifacts. Even so, biomonitoring investigations and scoping reviews have reported plastic particle detection in blood and several tissues, reinforcing the need for harmonized methodologies and shared quality criteria[15,16]. By contrast, polymeric nanoparticles used for drug delivery (e.g., PLGA- and PEG-based platforms and polymeric micelles) are deliberately manufactured with defined composition, tight size distributions, and adjustable surface features (such as

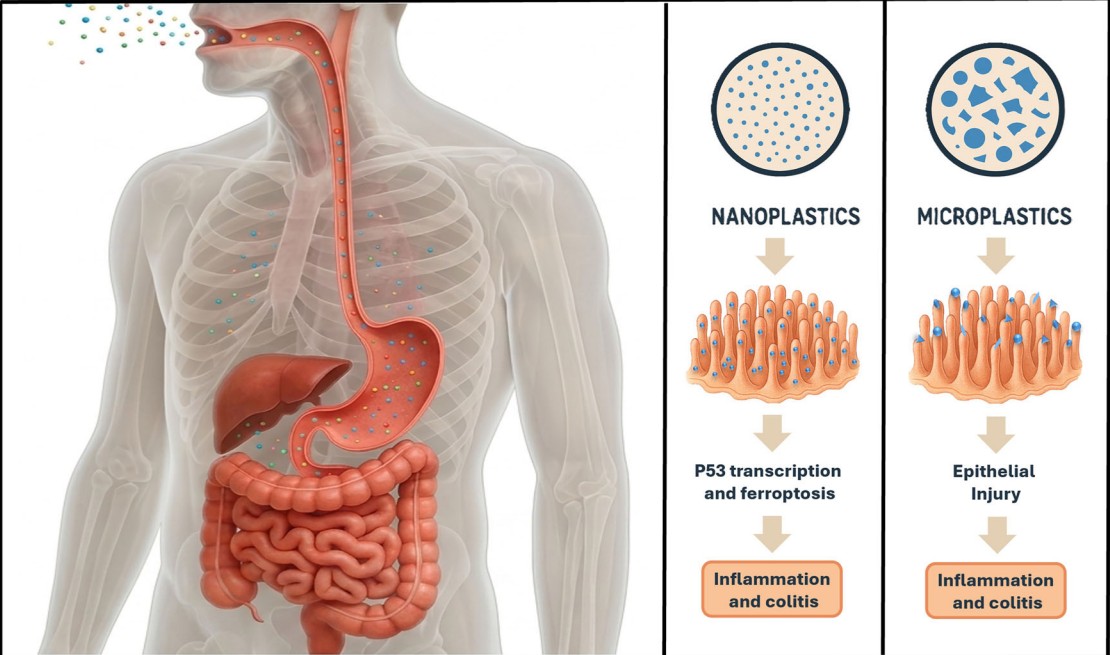

**Fig. 1 | Size-dependent mechanisms by which MNPs cause intestinal epithelial injury.** Adapted from [Cheng, Y., Chen, J., Fu, R. et al. Molecular mechanism differences between nanoplastics and microplastics in colon toxicity: nanoplastics induce ferroptosis-mediated immunogenic cell death, while microplastics cause cell metabolic reprogramming].

**Table 1 | Key differences between environmental MNPs and engineered nanoparticles**

| Property | Environmental MNPs (unintentional) | Engineered polymeric nanoparticles (intentional) |
|---|---|---|
| Origin[13] | Fragmentation/abrasion; secondary particles; mixed sources; environmental aging | Designed manufacturing for defined function |
| Composition[13–15] | Mixed polymers + additives; weathered/oxidized surfaces; adsorption of co-pollutants | Defined polymer/excipients; controlled residuals and degradation (e.g., PLGA hydrolysis); PEGylation as design feature |
| Size range & distribution[15] | Broad (nm–mm) and poorly standardized across studies | Typically narrow distributions (often tens–hundreds of nm) |
| Physicochemical predictability[13–15,19] | Low, due to aging, heterogeneity, and contaminant adsorption | High, due to controlled synthesis and quality-by-design manufacturing |
| Morphology[15] | Heterogeneous (fibers/fragments/films/spheres), irregular surfaces | Reproducible morphology (often spherical); controlled structure |
| Surface control[13,19] | Uncontrolled; oxidized/weathered; variable corona/biofilm; changing charge/hydrophilicity | Tunable (PEGylation, ligands, charge); reproducible |
| Contaminants burden[13,14,17] | High likelihood of adsorbed POPs/metals/PAHs + environmental bioburden | Low/controlled impurities; sterility/endotoxin testing in translational pipelines |
| Exposure context[15,17] | Chronic, involuntary; ingestion/inhalation; dose uncertain | Defined dose/route/schedule; toxicity monitoring |
| Evidence base[17] | Emerging; high heterogeneity; contamination bias; limited causal inference | Standardized preclinical packages; quality attributes/QbD emphasized for translation |
| Regulation[17,19] | Environmental pollutant; not regulated as a medicinal product | Regulated as drug/biologic/device depending on product; GMP/QC requirements |
| Intended function[19] | No intended biological function; potential hazard | Therapeutic function by design (delivery, targeting, controlled release) |

PEGylation and targeting ligands), enabling more reproducible pharmacokinetics and biodistribution. PLGA carriers, in particular, are extensively studied because PLGA is a biodegradable polymer with longstanding use in pharmaceutical technologies; multiple reviews stress that successful clinical translation depends on standardized toxicology packages, along with stringent control of critical quality attributes, including composition, residual solvents, endotoxin burden/sterility, stability, and release kinetics[17,18]. Notably, PEGylation, while commonly employed to extend circulation time, also carries clinically meaningful considerations (e.g., anti-PEG antibodies and hypersensitivity reactions). These issues are typically addressed within controlled development programs and regulatory evaluations, rather than representing an uncontrolled environmental modifier of hazard or exposure[19,20].

## Regulatory landscape and translational implications

A key obstacle to effective translation into policy and practice is the continuing legislative and regulatory shortfall, which in many ways reflects the scientific fragmentation of the field. Reported levels of microplastic contamination are often not comparable across geographic areas because sampling, detection, and analytical workflows are not harmonized; with a lack of alignment among national and international legal frameworks representing a structural barrier to coherent, impactful public policy[21]. From a regulatory perspective, existing instruments frequently capture only part of the overall exposure problem. For instance, the European Union has implemented a REACH restriction aimed at reducing releases of synthetic polymer microparticles when they are intentionally present as such or deliberately incorporated into mixtures (e.g., certain consumer and industrial products). While this measure is relevant for primary microplastics, it does not directly address the larger contribution from secondary microplastics generated by the breakdown of macroplastics and by diffuse sources such as tire and road wear, textile shedding, and other forms of abrasion and fragmentation [ECHA microplastics[22,23]]. In parallel, global governance remains in flux: negotiations toward a legally binding international instrument on plastic pollution are ongoing within UNEP's Intergovernmental Negotiating Committee, underscoring that a fully coordinated global approach is still being constructed[24].

Even though causal inference in humans remains incomplete, the regulatory gap has practical clinical relevance: in the absence of enforceable, harmonized exposure limits and routine standardized monitoring, prevention largely depends on upstream actions (product requirements,

improved waste handling, and emission controls) rather than on clinical interventions. Thus narrowing legislative inconsistencies and standardizing measurement are best viewed as prerequisites for credible risk assessment and for any future clinical recommendations[21].

## Human internal exposure (body burden), heterogeneity, and analytical limitations

Human biomonitoring investigations have described plastic particles in several biological matrices, including blood, urine, and placental tissue, supporting the notion that internal exposure can occur. At present, however, there is no clinically validated metric that reliably captures an individual "body burden," nor are biological reference values or thresholds available. As a result, existing data are insufficient to define human dose–response relationships or to propose carcinogenic risk cutoffs. For example, the first quantitative, mass-based assessment reporting plastics in human blood estimated an average summed concentration in the µg/mL range, but the cohort was small and cross-study comparisons remain difficult because protocols and reporting practices vary substantially[15,25]. Similarly, Raman-based studies have reported microplastics in urine, yet the evidence is still early-stage and highly dependent on analytical sensitivity and stringent contamination control[26]. Reports using Raman and FTIR approaches on placenta samples further illustrate strong method dependence: detection frequencies and the apparent size spectrum can shift markedly depending on whether workflows only interrogate particles larger than ~50 µm or, alternatively, apply higher-sensitivity Raman procedures that capture particles in the low-micrometer range[27–29].

Considerable interindividual variability is also expected. Exposure profiles differ with diet, drinking-water source, indoor air and occupational environments, smoking, socioeconomic conditions, and co-exposures common in industrialized settings. Moreover, host-related factors, such as intestinal permeability, renal clearance, and inflammatory status, may modulate translocation and retention. Consequently, the detection of plastics in human specimens should be interpreted primarily as evidence of internal exposure and biological plausibility, rather than as proof of a uniform, clinically interpretable, or actionable "toxic load."

A further barrier to translation is analytical bias driven by non-harmonized measurement pipelines. Vibrational spectroscopy (µFTIR and µRaman) can provide particle-resolved information (polymer identification, size, and morphology), but performance is strongly influenced by particle size and matrix effects. In practice, many FTIR imaging workflows

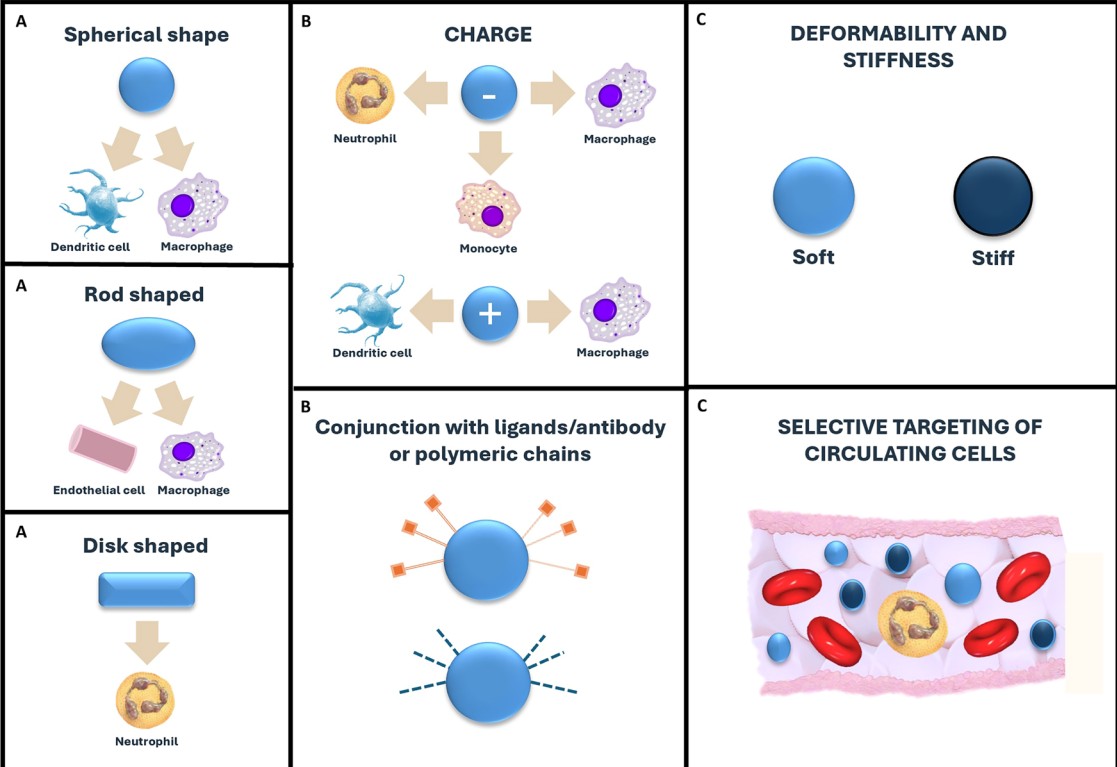

**Fig. 2 | Optimization of nanoplastics carriers. A** Nanoplastics can be formulated into a variety of shapes to target specific cell populations. **B** Functional groups on nanoplastics can be conjugated with targeting ligands or stealth polymeric chains. Additionally, nanoplastics can be positively or negatively charged. **C** The stiffness of the nanoplastics can be varied to target specific tumor cells in the bloodstream selectively.

have effective lower size limits around ~20 μm, whereas optimized μRaman/μFTIR methods can reach roughly ~1–5 μm; detection of smaller nanoplastics remains difficult and is often close to instrumental and procedural limits[30,31]. By contrast, pyrolysis-GC/MS offers robust, chemistry-specific, mass-based polymer quantification, yet it does not retain particle size or shape and therefore may not map cleanly onto particle-number metrics that could matter for barrier crossing and tissue distribution. Comparative work indicates that FTIR imaging and pyrolysis-GC/MS can yield divergent estimates depending on sample complexity and the units reported (counts versus mass)[32].

Overall, the field still lacks: (i) standardized protocols with rigorous contamination prevention, recovery assessment, and transparent quality reporting; (ii) harmonized reporting conventions (mass versus particle counts, size-class binning, and clearly stated limit of detection (LOD) / Limit of Quantification (LOQ); and (iii) well-designed prospective studies linking internal exposure measures to clinical endpoints while adequately addressing confounding. Therefore, although mechanistic findings and biomonitoring results support biological plausibility, it is premature to define clinical thresholds or to propose risk-stratified recommendations. Moreover, although a range of analytical methods is available to identify and characterize micro- and nanoplastics a widely accepted "gold standard" has yet to be defined.

Evidence on where micro- and nanoplastics localize at the subcellular level is derived mainly from cell-culture experiments and animal studies. In these models, particles are most often reported within the cytoplasmic compartment, frequently sequestered in endosomal and lysosomal vesicles, while nuclear proximity or entry has been described only in some conditions and appears to depend on parameters such as particle size, surface charge, and surface functionalization[33–35]. Although these observations are consistent with the biological plausibility of MNP-driven genotoxic effects, they cannot yet be translated straightforwardly to human genitourinary tissues. Indeed, direct and reproducible demonstrations of subcellular MNP

localization in human GU specimens are currently lacking, largely because of methodological constraints.

## Biocompatibility and toxic effects of micro-nanoplastics in the human body

Current evidence does not identify any specific lifestyle as definitively responsible for microplastic accumulation in the bladder or prostate. However, habits associated with higher overall exposure, such as frequent consumption of bottled water, heating food in plastic containers, regular intake of take-away food, and prolonged exposure to indoor dust and synthetic fibers, may plausibly increase the burden of microplastics in the urinary system. After entering the human body, MNPs tend to accumulate in various tissues due to their persistence and poor degradability[26,36,37].

The biocompatibility of MNPs, defined as their uptake, accumulation, and toxic effects in biological systems, is primarily determined by their size, shape, and surface properties (Fig. 2). The smaller the particle is, the more easily cells internalize it, resulting in higher accumulation rates[38].

Experimental studies suggest that exposure to micro- and nanoplastics can elicit a broad range of biological effects, including disruption of epithelial and endothelial barrier integrity, hormone-linked alterations, and inflammatory as well as oxidative stress–related outcomes[39]. Here, we focus on outlining the overall toxicological pattern and the principal organ systems reported to be affected[9]. Moreover, because MNPs can adsorb toxic chemicals (e.g., heavy metals, persistent organic pollutants) on their surface, they may serve as carriers of additional harmful agents, amplifying their cytotoxic and genotoxic effects.

A scoping review of 129 studies highlighted potential links between MNP exposure and neurotoxicity, metabolic disorders, and an elevated risk of cancer, underscoring their emerging role as a public health concern[40]. After inhalation or oral intake, experimental research points to the lung and liver as key organs in which MNP-associated toxicity may manifest, and it also reports secondary changes affecting intestinal barrier integrity and the

composition of the gut microbiota[41–43]. MNPs can also affect the reproductive system, with altered levels of testosterone, and reduced cell count and deformities in reproductive cells observed both in males and in females[44].

## Micro-nanoplastics and cancer risk development

At the cellular level, microplastics enter cells via clathrin- and caveolae-mediated endocytosis, by modulating intracellular signaling networks and, in experimental systems, aligning with shifts indicative of aberrant cell growth and other features commonly associated with carcinogenic transformation[45]. Many plastic-associated chemicals, such as vinyl chloride, dioxins, phthalates, and benzene, are classified as known or probable carcinogens[46]. Ecological studies have described geographic patterns in which elevated cancer rates coincide with regions burdened by greater plastic-derived pollution or presumed MNP contamination. These signals, however, should be treated as correlational rather than causal, given the likelihood of residual confounding (including co-occurring industrial emissions, socioeconomic differences, and lifestyle-related determinants) and the frequent imprecision of exposure characterization at the population level[47]. Alarmingly, a rise in both benign and malignant tumors has also been observed in other species, including sea turtles, sea lions, and Tasmanian devils, further supporting the potential oncogenic effects of MNPs pollution[48].

Evidence from in vitro and in vivo studies suggests that MNPs promote genotoxicity and DNA damage primarily through chronic inflammation and oxidative stress. These processes contribute to the development of a tumor microenvironment favorable for malignant transformation, consistent with inflammation being a key hallmark of cancer[49]. Several groups have tried to determine whether MNP exposure can trigger inflammation and impact immune system activity, measuring the levels of different cytokines involved in the inflammatory process, and in the regulation of innate or adaptive immune responses, such as Interleukin 1 (IL-1), IL-2, IL-6, IL-10, and Tumor necrosis factor alpha (TNF-α)[50,51]. In response to MNPs, an increased secretion of TNF-α, IL-1, and IL-10 was found, suggesting that innate immune cells, such as macrophages and dendritic cells, can recognize MNPs as pathogens and phagocytize them; therefore, MNPs are more likely to trigger innate than adaptive immunity. However, due to their size, MNPs are often difficult to remove and persist within the human body, starting the loop of chronic inflammation through the secretion of pro-inflammatory cytokines such as IL-6 or IL-8, ultimately altering the TME and, in the long term, inducing a shift of the immune system towards an immunosuppressive status[52,53].

In addition, exposure to micro- and nanoplastics has been linked to elevations in cellular oxidative stress, including increased reactive oxygen species production and disruption of redox homeostasis, which may in turn affect DNA integrity and overall genomic stability. Experimental models also implicate plastic-related additives in gene-expression shifts across pathways involved in inflammation, antioxidant and stress responses, proliferation and apoptosis, and regulation of the cell cycle, including signaling networks such as MAPK, Nrf2, PI3K/Akt, and TGF-β[53]. Further details related to various mechanisms through which MPs cause cancer development are explained in Table 2[49,54–68].

## MNPs and prostate cancer risk

Tumor and para-tumor samples were analysed from 22 patients with PCa after radical prostatectomy, providing quantitative and qualitative evidence of the presence of MNPs in tumor tissues through laser direct infrared spectroscopy, scanning electron microscopy, and pyrolysis-gas chromatography-mass spectrometry. MNPs were found in both para-tumor and tumor tissues of human prostate, with higher abundance within tumor tissues. Notably, there was a positive correlation between the abundance of MNPs and the frequency of consumption of take-out food[69]. Additives play a further role in cancer development in relation to MNPs, such as antioxidants and plasticizers, which can disrupt the androgen and estrogen pathways, leading to a higher risk of developing certain cancer subtypes, including PCa[70]. For

instance, the urine levels of bisphenols and phthalates levels, measured by an ultra-performance liquid chromatography-tandem mass spectrometry, were higher in 187 patients with PCa compared with 151 healthy controls, especially for some metabolites such as bisphenol A and di-isobutyl phthalate[71].

## MNPs and bladder cancer risk

Recently, MNPs have been investigated for their potential role in bladder cancer (BCa) onset. Indeed, the link between chemical exposure to aromatic amines, such as 2-naphthylamine, benzidine, 4-aminobiphenyl, and o-toluidine, and BCa has been widely demonstrated. Other chemicals, such as those used in the dye, textile, rubber, and paint industries, as well as cyclophosphamide and arsenic in drinking water, have also been investigated for their association with a higher risk of developing this tumor.

The presence of MNPs in 10 patients during bladder resection was demonstrated by comparing BCa and control cases through Raman microspectroscopy analysis with hyperspectral imaging performed in the range of 600–1800 cm$^{-1}$. In 13 out of 20 samples, PS-MPNs were detected. The risk of accumulation of MNPs in the bladder was proposed to be a possible factor in the development of cancer, with routine pathological and immunofluorescence tests being unable to detect their presence. The study was unable to answer the question of whether food contamination or ingestion was the cause. However, it was also emphasized that the type of spectroscopy used could potentially underestimate the amount of MNPs, due to the limitation in detecting particles smaller than hundreds of micrometers[72].

At the American Urology Association (AUA) 2025 congress, a study presented identified primary and secondary MNPs as being associated with a higher risk of BCa in the Ohio population, as observed over 8 years from 2013 to 2021. Application of Bivariate Moran's I for spatial correlation between waste release and cancer incidence, as well as geographic analysis and Local Indicators of Spatial Association, identified spatial clusters. A higher incidence of BCa was found across some areas with recognized plastic processing waste, associated with MNPs in water and air.

An exhaustive scoping review from 2021 provides the most accurate evidence regarding the presence of MNPs in urine samples from patients with GU tumors, primarily containing PS samples. This agent was able to compete with cisplatin, in cell models, for its cytotoxic and pro-apoptotic effects. Therefore, MNPs represent a threat to the human urinary tract cells by causing toxicity and inflammation, reducing cell survival and viability, and disrupting the MAPK (mitogen-activated protein kinase) pathway, which responds to growth signals and stress[73].

## MNPs and RCC risk

Several studies have also investigated the potential relationship between MNPs and RCC, especially for PS and acrylonitrile[74]. There is evidence of MNPs accumulating in kidney tissues, especially in kidney tubular cells (HK2), where they can exert different pro-tumorigenic effects, such as the production of ROS, the disruption of lipidic metabolism, endothelial reticulum-mediated stress damage, inflammation, fibrosis, and autophagy[75]. Furthermore, studies, including those focused on the kidney, suggest that MNPs may exert additive effects with environmental pollutants, potential carcinogens, and high-fat diets, ultimately modifying the tumor microenvironment through activation of the PI3K/Akt/MAPK and IL-17 signaling pathways in endothelial cells, effector CD8$^+$ T cells, and proliferating T cells[76].

Mechanistic research has repeatedly indicated that micro- and nano-plastic exposure can trigger oxidative imbalance, inflammatory signaling, genotoxicity-related outcomes, and disturbances of endocrine function; nevertheless, the body of evidence in humans is still comparatively sparse. Scientific literature is dominated by in vitro work, animal studies, and observational (including ecological) investigations, all of which have important limitations when it comes to drawing causal conclusions[77]. Ecological associations, in particular, can be distorted by factors common in highly industrialized settings, such as concurrent exposure to metals, PAHs, and aromatic amines, as well as by lifestyle and socioeconomic differences,

**Table 2 | Summary of carcinogenic mechanisms associated with micro- and nanoplastics**

| Mechanism | Key processes & effects | Link to tumor progression | Associated cancers / effects |
|---|---|---|---|
| Pro-inflammatory Response[52–54] | - Trigger immune response (macrophages, neutrophils).<br>- Lead to chronic inflammation due to persistence.<br>- Sustained release of IL-6, TNF-α.<br>- Generation of ROS (reactive oxygen species/RNS. | - Alters tumor microenvironment (TME).<br>- Promotes stromal remodeling, angiogenesis.<br>- Tumor-associated macrophages (TAMs) secrete VEGF.<br>- ROS/RNS cause DNA damage.<br>- Impairs immune surveillance, allowing tumor growth. | Liver Cancer, Lung Cancer, Colon Cancer |
| Oxidative Stress[55–57] | - Adsorb metal ions (Fe, Zn, Cu) that disrupt redox balance.<br>- Inhibit antioxidant enzymes.<br>- Increase intracellular ROS. | - ROS damage cell membranes, proteins, and DNA.<br>- Causes DNA strand breaks and mutations.<br>- Increased mutation rate drives uncontrolled cell proliferation. | General increased cancer risk |
| Endocrine Disruption & Hormonal Disruptions[58–60] | - Additives (plasticizers, antioxidants) and adsorbed Endocrine Disrupting Chemicals (EDCs) present in MP leach out.<br>- Phthalates and BPA disrupt estrogen/androgen function.<br>- Bisphenol A mimics thyroid hormones.<br>- EDCs bind to hormone receptors, mimicking or blocking natural hormones. | -Hormone imbalance triggers abnormal cell growth.<br>-Stimulates proliferation in hormone-responsive tissues.<br>-Initiates oxidative stress and inflammation. | Breast Cancer, Prostate Cancer, Ovarian Cancer, |
| Genotoxicity[49,61,62] | - Direct physical interaction with DNA and cellular components.<br>- ROS-induced DNA damage (strand breaks, fragmentation).<br>- Leaching of toxic additives (plasticizers, stabilizers).<br>- Induction of chromosomal abnormalities and spindle disruptions. | - Causes genetic instability and mutations.<br>- Disruption of cell division leads to aneuploidy (abnormal chromosome number).<br>- Collective DNA damage impairs cell function and increases cancer risk. | General increased cancer risk |
| Disruption of Cell Signaling Pathways[61,63] | - Endocrine-disrupting chemicals (EDCs) bind to hormone receptors.<br>- Imbalance key pathways: MAPK and PI3K/Akt.-Activate IRE1/XBP1 signaling axis of Endoplasmic Reticulum Stress (ERS) through oxidative stress<br>- Polypropylene activating the mTORC2 and AKT signaling pathways. | - Altered cellular responses lead to developmental problems and cancer.<br>- Expression of apoptotic markers, such as caspase-3 and caspase-12<br>- Disruption of cell cycle regulation and apoptosis (programmed cell death). | Kidney related Cancer, Breast Cancer |
| Carcinogenic Properties[49,64] | - Act as carriers for carcinogens (PAHs, heavy metals).<br>- Physical properties (size, shape, charge) aid cellular penetration and tissue accumulation.<br>- Leach carcinogenic additives (e.g., phthalates, flame retardants). | - Direct DNA damage and increased mutation rate.<br>- Promotes cell proliferation and suppresses apoptosis. | Bladder, Reproductive Cancers |
| Epigenetic Changes[65,66] | - Alterations in DNA methylation(hypo/hypermethylation).<br>- Histone modifications (acetylation, methylation).<br>- Dysregulation of non-coding RNAs (e.g., miRNAs). | - Silencing of tumor suppressor genes or activation of oncogenes.<br>- Altered gene expression in cell growth, apoptosis, and immune response. | Neurotoxicity, Cancer |

and by substantial uncertainty in measuring external exposure and estimating internal dose. Accordingly, here we focus on reported associations and on biological plausibility, while underscoring that a causal link cannot currently be established. Clarifying whether, and to what extent, MNPs influence cancer risk will require robust prospective cohort studies, more accurate exposure assessment at the individual level, and harmonized analytical protocols to detect and characterize MNPs in human biological matrices.

Establishing a direct causal role for micro- and nanoplastics (MNPs) in genitourinary (GU) cancers would require multiple, mutually reinforcing forms of evidence. From a human research perspective, the strongest support would come from large prospective cohort studies in which MNP exposure is quantified prior to cancer onset. Key requirements would include a demonstrable exposure–response pattern (greater internal exposure corresponding to higher risk) and the persistence of associations after stringent control for major confounders and correlated exposures, such as smoking, occupational contact with aromatic amines, arsenic, metals, PAHs, and socioeconomic and work-related factors. Beyond epidemiologic signals, the case for causality would be markedly strengthened by human-relevant mechanistic data indicating that MNPs are not merely detectable but functionally engaged within GU target tissues. This would entail showing that particles reach and interact with pertinent anatomical and cellular compartments (e.g., the urothelial interface, prostate tissue, and

renal tubular regions) and that increasing internal burden aligns with cancer-relevant intermediate endpoints, such as oxidative stress and DNA damage markers, persistent inflammatory activation with microenvironmental remodeling, endocrine-disruption profiles (particularly salient for prostate biology), and signatures of epithelial injury–repair or fibrotic remodeling (notably relevant to kidney and bladder)[78].

### Micro-nanoplastics and cancer therapy

Engineered micro- and nanoplastics (MNPs) are emerging as versatile platforms for drug delivery and cancer therapy, leveraging their small size, modifiable surfaces, and ability to cross biological barriers. They have been designed to ensure biocompatibility, targeting, and payload release, enabling a wide range of therapeutic applications. Below, we summarize some of their mechanisms of action, but we must keep in mind that environmental MNPs can also behave in a similar, yet uncontrolled, manner, potentially creating pro-tumor effects.

- **Changes in drug disposition (ADME):** Experimental work suggests that micro- and nanoplastics (MNPs) could, in principle, shift the pharmacokinetics of anticancer agents by modifying absorption, tissue distribution, metabolism, and elimination. This would influence both therapeutic activity and toxicity. Proposed explanations include effects at barrier interfaces that shape uptake and systemic availability, and

liver-mediated changes that could alter biotransformation capacity, circulating half-life, and drug persistence in blood[79–81].

- **Transporter- and uptake-related mechanisms linked to resistance phenotypes:** At the cellular level, MNPs have been hypothesized to affect how drugs enter and leave cells, for example by perturbing membrane transport processes and efflux activity that govern intracellular drug concentrations. If such shifts reduce intracellular exposure or promote adaptive stress responses, they could contribute to resistance-like outcomes in experimental settings, although this remains far from established in humans[79–81].

- **Protein corona formation, opsonization, and immune clearance:** After entering biological fluids, particulate materials can rapidly acquire a coating of adsorbed biomolecules, especially proteins, creating a "corona" that promotes opsonization and recognition by macrophages and other phagocytic cells, thereby shaping uptake and clearance kinetics[82,83]. In theory, repeated environmental exposure to MNPs could modulate innate immune handling of particulates or compete for clearance pathways, potentially influencing responses to other foreign particles, including therapeutic nanocarriers.

- **Tumor-directed delivery** via **controllable particle engineering:** The clinically relevant "nanomedicine" opportunity space largely involves pharmaceutical-grade polymer nanoparticles, not environmentally weathered plastics. These systems are deliberately engineered, through size control, surface chemistry, and optional ligand functionalization, to increase drug accumulation at tumor sites while reducing off-target exposure[79].

- **Prolonged local exposure and improved penetration for intravesical therapy:** For bladder cancer, polymer carriers are often designed to extend residence time within the bladder and enhance penetration into the urothelium, aiming to raise local drug levels without proportionally increasing systemic burden[84,85].

- **Immunotherapy support through targeted immunomodulator delivery:** Polymer platforms can be used to package and co-deliver immunomodulators or checkpoint-directed agents to immune cells or tumor-associated compartments, with the goal of strengthening antitumor immune activation and counteracting immunosuppressive features of the tumor microenvironment[86,87].

- **Imaging and image-guided intervention:** Engineered nanosystems can also function as targeted diagnostic probes, improving sensitivity and spatial precision for detection, prognostic assessment, and intraoperative guidance (e.g., receptor-targeted constructs in prostate cancer)[88,89].

Across these applications, real-world translation hinges on reproducible manufacturing and characterization, predictable biodistribution and elimination, careful management of immunogenicity (including anti-polymer/anti-PEG issues where relevant), and validation in robust in vivo and clinical studies rather than isolated proof-of-concept demonstrations[90,91].

## Conclusions

The evidence base on micro- and nanoplastics (MNPs) and genitourinary (GU) oncology is expanding quickly, yet it remains fragmentary and heterogeneous. For now, the most defensible interpretation of "risk" is not demonstrated causality, but a plausible biological rationale. In line with multidisciplinary evaluations of carcinogenic potential, interpretation should emphasize core causal requirements—temporality, dose metrics that reflect realistic internal exposure, and mechanistic models with high human relevance (including organoids and organs-on-chips)—rather than relying on isolated experimental readouts[92]. MNPs have been reported in human matrices connected to the GU system, and experimental work repeatedly identifies cancer-relevant perturbations, including redox imbalance, sustained inflammatory activation, endocrine disruption, and impairment of barrier function. Importantly, early tissue-based observations are emerging in GU oncology: microplastics have been detected in human GU tumor specimens, with higher particle burden and broader polymer profiles in malignant tissue than in matched adjacent non-malignant samples[93]. At the cellular scale, internalized nanoplastics may engage converging stress pathways. They can disrupt autophagic processing and proteostasis, impair mitochondrial function, and elicit endoplasmic reticulum stress. Together, these effects may promote ROS-associated inflammatory signaling. Potential genotoxic sequelae, including DNA damage, have also been discussed in relation to intracellular localization and cell-cycle context, especially in rapidly renewing epithelia[94]. Looking ahead, the principal bottlenecks are methodological and epidemiologic. The field needs harmonized analytical workflows with rigorous contamination prevention to support reproducible detection and quantification, followed by adequately powered prospective studies at the individual level that combine internal MNP measurements with established GU cancer determinants and key co-exposures to evaluate temporality, exposure–response patterns, and organ-specific vulnerability. In parallel, a developing clinical–epidemiological line of inquiry has proposed patient-oriented hypotheses for possible chronic effects, including a provisional "Microplastic Syndrome" construct that is explicitly exploratory and requires longitudinal verification[95]. By contrast, most of the near-term clinical opportunities discussed here arise from engineered polymeric nanomaterials manufactured under controlled specifications. In GU oncology, these platforms can address concrete translational challenges: improving intratumoral delivery, prolonging intravesical residence time, enabling immunomodulation within an immunosuppressive tumor milieu, and strengthening diagnostic accuracy and image-guided interventions. Overall, polymer-based nanoscale materials constitute a true "double-edged sword" in GU oncology, with ambient MNPs as an emerging exposure that warrants rigorous risk characterization and engineered polymeric nanoparticles as regulated diagnostic and therapeutic technologies. On the exposure side, progress depends on moving beyond qualitative detection toward quantitative internal dose measures and, ultimately, causal inference. Achieving this will require coordinated efforts spanning environmental science, analytical chemistry, urology, oncology, immunology, and materials engineering. Only by integrating methodological rigor with mechanistic coherence and translational realism can the field define potential risks linked to pervasive MNP exposure while responsibly leveraging engineered nanomaterials to improve outcomes in urogenital cancers.

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

## Acknowledgements

We would like to thank the ARON Research Foundation ETS for its valuable intellectual support and encouragement throughout the preparation of this manuscript. We wish to sincerely acknowledge DeepSeek for improving the language clarity and grammar in this manuscript.

## Author contributions

Conception and design: F.Mas., M.S., K.S., B.A.M.; acquisition of data: K.S., B.A.M.; analysis and interpretation of data: F.Mas., M.S., K.S., B.A.M.; drafting of the manuscript: F.Mas., M.S., K.S., B.A.M.; critical revision of the manuscript for important intellectual content: K.S., B.A.M., F.R., F.Mar., E.G., T.T., J.M.C., S.B., F.P., F.Mas., M.S.; supervision: F.Mas., M.S.

## Competing interests

The authors declare no competing interests.
