## [Transparent Peer Review file · Communications Medicine]

Microplastics as both a driver of genitourinary cancers and a deliverer of treatments

Corresponding Author: Professor Francesco Massari

Version 0:

Reviewer comments:

Reviewer #1

(Remarks to the Author)

The paper "The Double-Edged Sword: Microplastics as a Driver and a Deliverer in Genitourinary Cancers" address a highly novel, relevant, and rapidly expanding topic, integrating environmental, toxicological, and oncological evidence on micro- and nanoplastics (MNPs) in genitourinary (GU) cancers. The "double-edged sword" metaphor is well-suited to the content (risk vs. therapeutic opportunity). The authors discuss a very cutting-edge and relevant subject. The connection between MNPs and GU cancer is still in its infancy; from an excretory standpoint, the inclusion of the kidney, bladder, and prostate seems appropriate. The organization of the manuscript (exposure → biocompatibility → carcinogenesis → therapeutic impact → applications) is straightforward and progressive. Human studies (bladder, prostate) are included by the authors, which is unusual in this field. Conceptually, the difference between MNPs as pollutants and as therapeutic platforms is remarkable.

Although the topic and data-results presented seem appropriate for publishing, the authors are asked to refine the work for additional review. Below are my remarks:

1. The differences between synthetic polymeric nanoparticles (PLGA, PEG, micelles, and QDs) and unintentional ambient MNPs are not made explicit in the manuscript. By mistakenly implying toxicological or functional continuity, this could deceive the reader. A comparative table (origin, size, surface control, toxicity, regulation) and an explicit subsection (i.e., "Environmental MNPs vs. Engineered Polymeric Nanoparticles: fundamental differences") should be included.

2. Despite being a review, several sections of the paper imply that MNPs and cancer are causally related: Examples that are problematic include "higher cancer incidence in areas with high environmental presence of MNPs." "MNPs may be involved in the development of cancer." The majority of the evidence that is currently available comes from observations, in vitro experiments, or animal models. It is recommended that conditional language be improved (make a clear distinction between causality, biological plausibility, and association).

3. The manuscript lacks an analysis of the legislative gap regarding MNPs and their real-world clinical implications (screening, biomarkers, prevention). This limits its translational impact. Please review the proposed citation (DOI: 10.1007/s11270-024-07589-1). The article would support the environmental-clinical discussion by highlighting the need for public policies and introducing the regulatory gap as a structural issue.

4. The manuscript lacks clinical and epidemiological integration. Isolated studies in human tissues are cited, but the body burden, biological thresholds, interindividual heterogeneity, and detection biases (i.e., FTIR, Raman, pyrolysis) are not discussed. I advise the authors to provide information on these issues.

5. The manuscript presents redundancies: Repetition of mechanisms (ROS, inflammation, MAPK) in sections 2 and 3. Some references are conceptually duplicated.

6. Citing the following listed articles would substantially improve the scientific, conceptual, and clinical quality of the manuscript:

- DOI: 10.1002/jat.4598. This article would help the authors delve deeper into intracellular localization and genotoxicity. Additionally, it provides advanced molecular mechanisms (i.e., DNA damage, nuclear stress). The authors could extract

useful information for sections 2 and 3 of their manuscript.

- DOI: 10.1016/j.envpol.2025.127343. This article provides a critical analysis of the criteria for carcinogenicity. It offers a broad conceptual framework and aids in moderating causal judgments. Authors would benefit from this article by avoiding overinterpretations.
- DOI: 10.3390/microplastics4040093. A clinical-systemic perspective is offered in this reference. It links actual clinical symptoms to long-term exposure. To increase the manuscript's medicinal impact, the authors could extract relevant information. For a clinical oncology audience, it is very useful.
- DOI: 10.1016/j.scitotenv.2025.178815. Human GU tumor tissue is used as direct evidence in this reference. The anatomical spectrum is extended beyond the bladder and prostate. The authors would benefit from this article by strengthening the biological plausibility, especially in the part on GU cancers.

7. In order helping authors in enhancing the scientific quality of their manuscript, the following inquiries are provided:

- How can authors compare studies that use FTIR, Raman, SEM-EDS, and pyrolysis-GC/MS?
- What kind of evidence do you suppose is required to prove that MNPs directly cause GU cancer?
- Do human GU tissues have direct data on subcellular localization (nucleus vs. cytoplasm)?
- Which technique ought to be regarded as the gold standard for finding MNPs in human tissues?
- Could outdoors MNPs cause opsonization or competition with medicinal nanoparticles?
- Do you think it's possible to employ MNPs as biomarkers of exposure or cancer risk?

The manuscript would reach a high degree of scientific rigor, conceptual clarity, and clinical relevance with the recommended changes, making it entirely appropriate for publication. Its influence and translational value would be significantly increased by the modifications recommended.

Reviewer #2

(Remarks to the Author)

The topic is timely and addresses the emerging issue of micro- and nanoplastics (MNPs) in human health, specifically their dual role in genitourinary (GU) cancers. While the authors have compiled a broad overview of the potential links between MNPs and carcinogenesis, as well as their potential therapeutic applications, the manuscript in its current form lacks the scientific rigor, clarity, and novelty required for publication in a reputable journal. Below are my major concerns:

Major Concerns:

1. The manuscript presents itself as a review, but it largely aggregates previously published findings without offering a critical, integrative, or forward-looking perspective. The central premise—MNPs as both a risk factor and a therapeutic tool—is interesting but is not explored in sufficient depth. The sections on therapeutic applications, in particular, read like a list of examples rather than a coherent analysis of mechanisms, challenges, or translational potential.
2. The authors repeatedly state that evidence is limited or insufficient to establish a direct causal link between MNPs and GU malignancies (e.g., lines 26–28, 70–71). Despite this, the manuscript often uses language that implies stronger associations than the data support. For example, the correlation between MNP abundance and take-out food consumption in prostate tumors (lines 125–134) is presented as compelling but remains observational and preliminary. Without a clear critical appraisal of the quality and limitations of these studies, the review may inadvertently overstate the risk.
3. The manuscript jumps between exposure pathways, toxicology, cancer risk, and therapeutic applications without clear transitional logic. Sections 3.1–3.3 (on prostate, bladder, and RCC) are brief and descriptive, failing to synthesize cross-cutting mechanisms or highlight organ-specific vulnerabilities. The therapeutic section (Section 4) is overly broad, covering everything from drug delivery to imaging probes without a unifying narrative.
4. A review should not only summarize but also evaluate the strength of evidence, identify knowledge gaps, and propose future directions. This manuscript does little to critique methodological limitations (e.g., detection limits of spectroscopy, heterogeneity of MNP compositions) or to contextualize findings within the broader landscape of environmental oncology. The conclusion calls for more research but lacks specific, actionable recommendations.
5. Several recent reviews have already covered the role of MNPs in cancer risk and nanomedicine applications. This manuscript does not clearly distinguish itself from prior work, nor does it offer a unique conceptual framework or novel insights that would justify publication.
6. References to figures and tables are occasionally unclear (e.g., Table I is mentioned but not provided in the submitted text). Some sections contain overly long sentences and jargon that impede readability. The abstract and conclusion are repetitive and do not succinctly capture the review's contributions.

Minor Points:

- The introduction could be more focused; it currently covers too much general background on MNP pollution.
- The “dual role” concept is not consistently developed throughout the manuscript; the two roles feel disconnected.
- Some statements are speculative and should be tempered with more cautious language (e.g., lines 229–234).

Overall Assessment:

While the topic is of significant interest and relevance, the manuscript in its current form does not meet the standards for a review article in a high-impact journal. It lacks a critical analytical perspective, coherent structure, and novel synthesis. The authors may consider a substantial rewrite that:

1. Clearly defines the scope and unique contribution of the review.
2. Provides a rigorous, critical evaluation of the evidence linking MNPs to GU cancers.
3. Develops the “double-edged sword” theme more thoughtfully, perhaps by comparing mechanisms of toxicity vs.

therapeutic design principles.

4. Identifies specific, unresolved questions and proposes a research agenda.

Given the extent of revision required, I recommend rejection. However, if the authors are willing to undertake a major restructuring and deepen the analytical rigor, a resubmission could be considered after external peer review.

Version 1:

Reviewer comments:

Reviewer #1

(Remarks to the Author)

Dear Editor,

Dear authors,

After making the necessary changes based on my suggestions, I believe the paper "Microplastics as both a driver of genitourinary cancers and a deliverer of treatments" is ready for publication in your prestigious journal.

Reviewer #2

(Remarks to the Author)

This review explores the dual role of micro- and nanoplastics (MNPs) in genitourinary (GU) oncology, presenting them as both emerging risk factors and potential therapeutic tools. It highlights the detection of MNPs in human prostate and bladder tumors, with higher abundances linked to dietary habits like take-out food consumption, and discusses possible carcinogenic mechanisms including chronic inflammation, oxidative stress, genotoxicity, and endocrine disruption. The article also emphasizes a critical distinction between unintentional environmental MNPs and engineered polymeric nanoparticles (such as PLGA and PEG-based carriers), which are being deliberately designed to enhance targeted drug delivery, improve intravesical therapy, and support immunotherapy in GU cancers. Ultimately, the authors call for standardized analytical methods and robust prospective studies to establish causal relationships, while acknowledging the translational potential of engineered nanomaterials in improving cancer treatment outcomes. However, there is some questions that need to be further explained.

1. What are the key differences between environmental microplastics and engineered polymeric nanoparticles in terms of origin, physicochemical properties, and regulatory status as discussed in the article?
2. The article mentions that microplastics have been detected in human prostate and bladder tumors. What correlation was found between the abundance of these particles and patient lifestyle factors?
3. How might environmental MNPs potentially interfere with the efficacy of anticancer drugs, and what role could they play in promoting drug resistance?
4. According to the article, what specific effects do MNPs have on the immune system and the tumor microenvironment (TME) that could promote tumor progression?
5. What future research directions and methodological improvements do the authors suggest are necessary to move from detecting MNPs to proving a causal role in genitourinary cancers?

Version 2:

Reviewer comments:

Reviewer #2

(Remarks to the Author)

I have no further comments or revisions at this point, and I agree that the manuscript can be accepted.

Dear Editor,

We've just submitted the Revised version of our manuscript entitled “**Microplastics as both a driver of genitourinary cancers and a deliverer of Treatments**”. We have completely markedly revised all the structure of the manuscript by adding new Sections and a new Table following all the requests by the Reviewers.

Thank you so much for the opportunity to collaborate with you and your Journal

Best regards

Prof. Francesco Massari; Medical Oncology, IRCCS Azienda Ospedaliero-Universitaria di

Bologna, Via Albertoni n.15, 40138 Bologna, Italy.

List of Corrections

Reviewer #1 (Remarks to the Author):

The paper "The Double-Edged Sword: Microplastics as a Driver and a Deliverer in Genitourinary Cancers" address a highly novel, relevant, and rapidly expanding topic, integrating environmental, toxicological, and oncological evidence on micro- and nanoplastics (MNPs) in genitourinary (GU) cancers. The “double-edged sword” metaphor is well-suited to the content (risk vs. therapeutic opportunity). The authors discuss a very cutting-edge and relevant subject. The connection between MNPs and GU cancer is still in its infancy; from an excretory standpoint, the inclusion of the kidney, bladder, and prostate seems appropriate. The organization of the manuscript (exposure → biocompatibility → carcinogenesis → therapeutic impact → applications) is straightforward and progressive. Human studies (bladder, prostate) are included by the authors, which is unusual in this field. Conceptually, the difference between MNPs as pollutants and as therapeutic platforms is remarkable.

Reply: We express our sincere thanks to the reviewer for his valuable time and suggestions on this manuscript.

1. The differences between synthetic polymeric nanoparticles (PLGA, PEG, micelles, and QDs) and unintentional ambient MNPs are not made explicit in the manuscript. By mistakenly implying toxicological or functional continuity, this could deceive the reader. A comparative table (origin, size, surface control, toxicity, regulation) and an explicit subsection (i.e.,

"Environmental MNPs vs. Engineered Polymeric Nanoparticles: fundamental differences") should be included.

Reply: We thank the reviewer for this important comment. We acknowledge that, in the original version of the manuscript, this distinction was not made sufficiently explicit. To address this concern, we have implemented the following revisions:

- We have added a dedicated subsection entitled “Environmental MNPs versus Engineered Polymeric Nanoparticles: Fundamental Differences”, explicitly clarifying that the therapeutic applications discussed refer exclusively to intentionally designed, biomedical-grade polymeric nanoparticles and not to unintentional environmental MNPs.
- We have included a comparative table [Table 1] summarizing the key differences between environmental MNPs and engineered polymeric nanoparticles with respect to origin, size distribution, surface control, purity, toxicological profile, biological behavior, and regulatory status.

Query 2. Despite being a review, several sections of the paper imply that MNPs and cancer are causally related: Examples that are problematic include "higher cancer incidence in areas with high environmental presence of MNPs." "MNPs may be involved in the development of cancer." The majority of the evidence that is currently available comes from observations, in vitro experiments, or animal models. It is recommended that conditional language be improved (make a clear distinction between causality, biological plausibility, and association).

Reply: We thank the reviewer for this comment. We acknowledge that some statements in the original version of the manuscript may have unintentionally implied a causal link. To address this concern, we have revised the manuscript to clearly distinguish between association, biological plausibility, and causation.

In particular, we added a section of text to Section 6.3, “MNPs and RCC risk.

Query 3. The manuscript lacks an analysis of the legislative gap regarding MNPs and their real-world clinical implications (screening, biomarkers, prevention). This limits its translational impact. Please review the proposed citation (DOI: 10.1007/s11270-024-07589-1). The article would support the environmental-clinical discussion by highlighting the need for public policies and introducing the regulatory gap as a structural issue.

Reply: We thank the reviewer for this suggestion. Now, we have added a brief paragraph in the manuscript addressing the legislative and regulatory gaps related to environmental micro- and nanoplastics (MNPs) also incorporating the suggested reference. In particular, we added the section “3. Regulatory landscape and translational implications: from legislative gaps to clinical relevance.”

Query 4. The manuscript lacks clinical and epidemiological integration. Isolated studies in human tissues are cited, but the body burden, biological thresholds, interindividual heterogeneity, and detection biases (i.e., FTIR, Raman, pyrolysis) are not discussed. I advise the authors to provide information on these issues.

Reply: We thank the reviewer for this suggestion. Now, we have expanded the manuscript to include a discussion of clinical and epidemiological considerations related to micro- and nanoplastics (MNPs) in humans. In particular, we added the section “4. Human internal exposure (body burden), heterogeneity, and analytical limitations.”

Query 5. The manuscript presents redundancies: Repetition of mechanisms (ROS, inflammation, MAPK) in sections 2 and 3. Some references are conceptually duplicated.

Reply: We streamlined some parts of the manuscript to reduce redundancies. In particular, at the sections “5. Micro-nanoplastics in the human body: biocompatibility and toxic effects” and “6. Micro-nanoplastics and cancer risk development”

Query 6. Citing the following listed articles would substantially improve the scientific, conceptual, and clinical quality of the manuscript:

Reply: We thank the reviewer for bringing these manuscripts to our attention, and we have incorporated them into our work.

Query 7. In order helping authors in enhancing the scientific quality of their manuscript, the following inquiries are provided:

- How can authors compare studies that use FTIR, Raman, SEM-EDS, and pyrolysis-GC/MS?

Reply: The paragraph “Human internal exposure (body burden), heterogeneity, and analytical limitations” that we added to the manuscript also contains comparisons between the techniques and some considerations.

- What kind of evidence do you suppose is required to prove that MNPs directly cause GU cancer?

Reply: To address this topic, we added text to the section “6.3 MNPs and RCC risk.”

- Do human GU tissues have direct data on subcellular localization (nucleus vs. cytoplasm)?

Reply: To address this topic, we added text at the end of the section “4. Human internal exposure (body burden), heterogeneity, and analytical limitations.”

- Which technique ought to be regarded as the gold standard for finding MNPs in human tissues?

Reply: To address this topic, we added text at the end of the section “4. Human internal exposure (body burden), heterogeneity, and analytical limitations.”

- Could outdoors MNPs cause opsonization or competition with medicinal nanoparticles?

Reply: We discussed this in the section “Protein corona formation, opsonization, and immune clearance,” within the paragraph “7. Micro-nanoplastics and cancer therapy.”

- Do you think it's possible to employ MNPs as biomarkers of exposure or cancer risk?

Reply: We discussed this in the “Introduction” section.

The manuscript would reach a high degree of scientific rigor, conceptual clarity, and clinical relevance with the recommended changes, making it entirely appropriate for publication. Its influence and translational value would be significantly increased by the modifications recommended.

Reply: We hope we have addressed all of the reviewer’s concerns.

Reviewer #2 (Remarks to the Author):

The topic is timely and addresses the emerging issue of micro- and nanoplastics (MNPs) in human health, specifically their dual role in genitourinary (GU) cancers. While the authors have compiled a broad overview of the potential links between MNPs and carcinogenesis, as well as their potential therapeutic applications, the manuscript in its current form lacks the scientific rigor, clarity, and novelty required for publication in a reputable journal. Below are my major concerns:

Reply: We express our sincere thanks to the reviewer for his valuable time and suggestions on this manuscript.

Major Concerns:

Query 1. The manuscript presents itself as a review, but it largely aggregates previously published findings without offering a critical, integrative, or forward-looking perspective. The central premise—MNPs as both a risk factor and a therapeutic tool—is interesting but is not explored in sufficient depth. The sections on therapeutic applications, in particular, read like a list of examples rather than a coherent analysis of mechanisms, challenges, or translational potential.

Reply: We thank the reviewer for this comment. We rewrote the section, which is now titled “7. Micro-nanoplastics and cancer therapy,” and we highlighted the mechanisms of action in bullet points.

Query 2. The authors repeatedly state that evidence is limited or insufficient to establish a direct causal link between MNPs and GU malignancies (e.g., lines 26–28, 70–71). Despite this, the manuscript often uses language that implies stronger associations than the data support. For example, the correlation between MNP abundance and take-out food consumption in prostate tumors (lines 125–134) is presented as compelling but remains observational and preliminary. Without a clear critical appraisal of the quality and limitations of these studies, the review may inadvertently overstate the risk.

Reply: We thank the reviewer for this comment. We acknowledge that some statements in the original version of the manuscript may have unintentionally implied a causal link. To address this concern, we have revised the manuscript to clearly distinguish between association, biological plausibility, and causation. In particular, we added a section of text to Section 6.3, “MNPs and RCC risk.

Query 3. The manuscript jumps between exposure pathways, toxicology, cancer risk, and therapeutic applications without clear transitional logic. Sections 3.1–3.3 (on prostate, bladder, and RCC) are brief and descriptive, failing to synthesize cross-cutting mechanisms or highlight organ-specific vulnerabilities. The therapeutic section (Section 4) is overly broad, covering everything from drug delivery to imaging probes without a unifying narrative.

Reply: We revised multiple parts of the manuscript, reorganized some sections, added new ones, included a table, and incorporated additional considerations.

Query 4. A review should not only summarize but also evaluate the strength of evidence, identify knowledge gaps, and propose future directions. This manuscript does little to critique methodological limitations (e.g., detection limits of spectroscopy, heterogeneity of MNP compositions) or to contextualize findings within the broader landscape of environmental oncology. The conclusion calls for more research but lacks specific, actionable recommendations.

Reply: To address these topics, we added the section “4. Human internal exposure (body burden), heterogeneity, and analytical limitations.”. In addition, we rewrote the “Conclusions” section.

Query 5. Several recent reviews have already covered the role of MNPs in cancer risk and nanomedicine applications. This manuscript does not clearly distinguish itself from prior work, nor does it offer a unique conceptual framework or novel insights that would justify publication.

Reply: The manuscript has now been substantially revised, and it now includes eight sections instead of the four in the previous version.

Query 6. References to figures and tables are occasionally unclear (e.g., Table I is mentioned but not provided in the submitted text). Some sections contain overly long sentences and jargon that impede readability. The abstract and conclusion are repetitive and do not succinctly capture the review’s contributions.

Reply: We added a table (Table 1), which is cited in the section “2. Environmental MNPs versus Engineered Polymeric Nanoparticles: Fundamental Differences.” Table 2 is cited in the section “6. Micro-nanoplastics and cancer risk development.”. Tables are located after the "References" section. We also identified redundancy in some parts of the manuscript, which we have now removed. The “Conclusions” section has been rewritten.

- The introduction could be more focused; it currently covers too much general background on MNP pollution.

Reply: Corrected as requested by the Reviewer.

- The “dual role” concept is not consistently developed throughout the manuscript; the two roles feel disconnected.

Reply: Corrected as requested by the Reviewer by improving each Section and adding new Sections to underline the dual role.

- Some statements are speculative and should be tempered with more cautious language (e.g., lines 229–234).

Reply: Corrected as requested by the Reviewer.

Point by point response

To the Editor

Thank you for your review. We've trimmed the abstract to 150 words, and a brief summary is provided below.

Short summary:

Sridharan et al. discuss the impact of microplastics on genitourinary cancers, distinguishing between environmental and man-made contaminants. Both potential risk factors and new therapeutic avenues in genitourinary oncology are highlighted.

To Reviewer #2

We sincerely thank the Reviewer for the time and effort devoted to the careful evaluation of our manuscript. We greatly appreciate the insightful comments and suggestions provided, which have helped us to improve the quality and clarity of the paper. We hope that the revisions we have made have strengthened the manuscript and that the changes are satisfactory to the Reviewer.

1. What are the key differences between environmental microplastics and engineered polymeric nanoparticles in terms of origin, physicochemical properties, and regulatory status as discussed in the article?

Authors: We thank the Reviewer for this comment. In response, we have revised Section 2, entitled "Environmental MNPs versus Engineered Polymeric Nanoparticles: Fundamental Differences," to provide a clearer and more structured discussion of these aspects. In addition, Table 1 has been updated.

2. The article mentions that microplastics have been detected in human prostate and bladder tumors. What correlation was found between the abundance of these particles and patient lifestyle factors?

Authors: We have added some lines to Section 5, "Micro-nanoplastics in the human body: biocompatibility and toxic effects," to better report the correlations between the abundance of microplastics and patient lifestyle factors.

3. How might environmental MNPs potentially interfere with the efficacy of anticancer drugs, and what role could they play in promoting drug resistance?

Authors: We revised Section 7, "Micro-nanoplastics and cancer therapy," to clarify that some mechanisms already described for engineered MNPs may also be applicable to environmental MNPs. These include, for example, the adsorption/sequestration of drugs and the alteration of cellular membrane transport processes, which may affect drug bioavailability, intracellular uptake, and ultimately therapeutic efficacy.

4. According to the article, what specific effects do MNPs have on the immune system and the tumor microenvironment (TME) that could promote tumor progression?

Authors: In Sections 6, “Micro-nanoplastics and cancer risk development,” and 6.3, “MNPs and RCC risk.” we explain how MNPs may affect the immune system and the tumor microenvironment by promoting chronic inflammation, modulating cytokine release, and activating signaling pathways involved in tumor progression.

5. What future research directions and methodological improvements do the authors suggest are necessary to move from detecting MNPs to proving a causal role in genitourinary cancers?

Author: In the Conclusions section we emphasize the key research directions and methodological improvements needed to move from the simple detection of MNPs toward demonstrating a causal role in genitourinary cancers. In particular, we highlight the need for rigorous and well-designed studies with adequate statistical power, careful consideration of additional confounding variables, and more standardized and reliable approaches to quantify the internal accumulation of MNPs in human tissues.